# Assessment and Monitoring of Local Climate Regulation in Cities by Green Infrastructure—A National Ecosystem Service Indicator for Germany

Ralf-Uwe Syrbe [1],*, Sophie Meier [1], Michelle Moyzes [2], Claudia Dworczyk [1] and Karsten Grunewald [1]

1 Leibniz Institute of Ecological Urban and Regional Development (IOER), Weberplatz 1, 01217 Dresden, Germany; s.meier@ioer.de (S.M.); c.dworczyk@ioer.de (C.D.); k.grunewald@ioer.de (K.G.)
2 Energiedienst Holding AG, Baslerstrasse 44, 5080 Laufenburg, Switzerland; michelle.moyzes@energiedienst.de
* Correspondence: r.syrbe@ioer.de; Tel.: +49-351-4679219

**Abstract:** In densely built-up urban areas, green spaces such as gardens, parks, forests and water bodies can greatly enhance the quality of life for local residents and promote human health. These areas mitigate heat stress and the urban heat island effect to create a balanced local climate. To quantify the ecosystem service of "urban climate regulation" provided by urban green infrastructure, we developed a national indicator for specific measurement and monitoring. This indicator captures both the supply of climate-regulating services by urban green spaces and the demand for this service from the residential population. Using nationwide geodata, a cooling capacity value can be calculated that reflects the tree canopy, soil cover, sizes of green area and site characteristics. This cooling capacity value is then related to the affected residential population in the neighbourhood. Our analysis indicates that 76% of the population in the 165 case cities in Germany enjoy high or very high cooling capacities in their immediate living environment. In 37 cities, over 85% of the population benefits from good or very good cooling capacity provided by green space. The proposed indicator enables a comparison of the cooling service of urban green infrastructure and offers a sound basis for spatial planning and decision-making in urban areas.

**Keywords:** climate adaptation; cooling; green space; urban heat island; urban planning

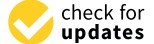



## 1. Introduction

The latest report of the Intergovernmental Panel on Climate Change (IPCC) highlights the global climate changes and impacts of extreme climate events such as hot summers, heatwaves, droughts and extreme rainfall [1]. Throughout Europe, alterations in various climatic impact-drivers have been observed in all regions. The IPCC has predicted ongoing significant climate warming and an increase in the frequency, duration and intensity of climate-related hazards by the year 2100 [2]. Heatwaves and hot summers are major threats in Europe, with an estimated 60,000-plus heat-related deaths in the summer of 2022 alone [3]. Periods of extreme heat can be particularly dangerous in urban areas due to the urban heat island (UHI) effect [4–6], which has an adverse impact on human health, livelihoods and infrastructure [2].

International efforts to mitigate climate change can be seen in the global Paris Agreement on climate change [7] or the European Green Deal [8], which is a comprehensive strategy involving various goals, including the restoration of biodiversity and mitigation of climate change, through the ambitious target of achieving net-zero emissions of greenhouse gases by 2050. In 2021, the European Green Deal was complemented by the EU Adaptation Strategy [9], which emphasises the importance of smarter, swifter and more systemic adaptation measures to combat the adverse impacts of climate change that we already face today [9].

Urban green spaces such as parks, gardens, lawns, street trees, urban forests, roof greening and water bodies play a crucial role in mitigating the UHI effect and regulating local climates in cities [10]. These green spaces can represent strategically planned networks of natural and semi-natural landscape elements—also referred to as urban green infrastructure (UGI) [11]—and provide essential ecosystem services, including local climate regulation [12]. By reducing insolation through shade, enhancing evapotranspiration (transpiration of plants as well as evaporation from moist soils and water surfaces) [4,13] and modifying thermal surface characteristics (e.g., radiation, heat storage), urban green spaces can effectively mitigate heat stress and thereby contribute to the overall well-being of urban populations [14,15].

As air temperatures rise and extreme weather events become ever more frequent and intense, the role of UGI in cooling and enhancing the local climate is increasingly seen as crucial [16]. However, robust data and information are needed to effectively prioritise and integrate UGI in urban planning and decision-making processes as essential climate adaptation measures and options for action [9,17].

There exist various approaches to assessing the benefits of UGI in local climate regulation, including physical models [18], expert-based assessments [19] and statistical models that simulate climate change effects [20]. However, it can be both challenging and time consuming to disentangle the various processes involved in cooling, especially under diverse weather conditions [21].

This article aims to address this challenge by presenting a national indicator for the ecosystem service of "local climate regulation in cities". Building on the existing Climate Cooling Assessment (CCA) approach of Zardo et al. [22], we created an adapted procedure to quantify the cooling capacities of German cities at the national scale. The original approach, designed specifically for daytime weather conditions with limited airflow, requires only a minimal set of input parameters, facilitating its application across larger geographical areas.

Using data on soil cover (land use and land cover as well as vegetation), the adapted CCA approach estimates specific cooling values tailored to hot summer days with low cloud cover, aligning with the climatic characteristics of various European climatic regions. The adoption of the CCA methodology as a foundation for this national indicator ensures compatibility with European data while incorporating finer national-level data, which has proved useful for other ecosystem indicators previously developed by the authors [14,23].

Moreover, the proposed indicator aligns with the objectives outlined in the European Biodiversity Strategy 2020 [11], reflecting the commitment of EU member states to assess the status and services of ecosystems and to integrate the findings into European and national reporting systems (Target 2 Action 5). In accordance with the requirements of the EU Biodiversity Strategy 2020, a system for the initial national assessment of ecosystem services has already been developed and harmonised for Germany [24]. This article contributes to this broader strategy by presenting a national indicator for the ecosystem service of "local climate regulation in cities", along with the methodology of its calculation and preliminary results for larger cities in Germany.

By combining national data on land use and land cover, vegetation and population density, we hope to provide a methodology to assess the cooling benefits of urban green infrastructure across different urban settings. In addition to assessing the cooling capacity of UGI, we also consider its social implications. By integrating population data, the proposed indicator reflects the demand for cooling services in urban areas, providing insights into the distribution of climate adaptation benefits. Furthermore, by highlighting the role of UGI in enhancing urban resilience and promoting human health, we seek to inform policy and planning efforts aimed at fostering sustainable and climate-resilient cities [25].

## 2. Materials and Methods

### 2.1. Study Area and Input Data

2.1.1. Study Area

The study was initiated by the MAES programme and is a contribution to the German national ecosystem reporting system [11] aimed at meeting EU standards. The working group MAES (Mapping and Assessment of Ecosystems and their Services) was established to provide information on progress in achieving Target 2 Action 5 of the EU Biodiversity Strategy in EU countries [11,26]. Clearly, for any nationwide system of ecosystem reporting, proposed indicators must be applicable on a national scale. We decided to test the new national indicator on large cities, which are most likely to be affected by the urban heat island effect and heat stress [6]. Large cities ([27,28]) were selected by the classification of a functional urban area (FUA). A FUA consists of a densely inhabited city with at least 50,000 inhabitants (plus their commuting zone), whose labour market is highly integrated into the city [27]. In summary, we analysed German cities with at least 50,000 inhabitants that are located within an FUA.

2.1.2. Input Data

To apply the Climate Cooling Assessment (CCA) approach [22] to Germany, we first had to select the best-possible data sources (preferably from a public institution) that (1) were available for the entire country; (2) were updated regularly; (3) had the highest resolution; and (4) showed a high level of accuracy. In addition, we drew on experiences from former projects to select and refine the data. Table 1 shows the data sources used. In part these are national datasets and in part regional datasets, with the latter used to calibrate the former.

**Table 1.** Data sources used to map, assess and calibrate the cooling capacity.

| Data Name and Source | Time Period | Available Information | Spatial Scale |
|---|---|---|---|
| Land cover model Germany LBM-DE [29] | 2018 | Polygons: area size Land/soil cover | Minimum mapping unit 0.2–1 ha |
| Cities > 50,000 inhabitants [30] | 2011 | City and IDs | 1: 25,000 |
| Administrative areas VG25 [31] | 2016 | Polygons: city boundaries | 1: 25,000 |
| Functional Urban Areas (FUA) of Urban Atlas [32] | 2018 | Polygons of the FUA areas | |
| Street Tree Layer of Urban Atlas (STL) [32] | 2018 | Polygon data of tree coverage (trees outside the forest) | 10 m × 10 m |
| Urban Green Raster Germany (UGR) [33] | 2018 | Raster data of tree coverage classes | 10 m × 10 m |
| Green volume data for 4 cities [34,35] | 2017/18 | Raster data of green leave area for calibration | 0.5 m × 0.5 m |
| Census: number of inhabitants [36] | 2011 | Raster distribution data Resident population | 100 m × 100 m |

The basic dataset for land cover or soil cover and the selection of urban green infrastructure was the German Land Cover Model (LBM-DE), developed by the Federal Agency for Cartography and Geodesy. Updated every three years, the LBM-DE is based on the German topographic model ATKIS Basis-DLM and Sentinel satellite data [29] (Table 1, line 1). The LBM-DE is used to derive the CORINE land cover (CLC) dataset for Germany, which includes a standardised EU definition for land cover and land use [29]. The LBM-DE is, however, more detailed in spatial scale, more frequently updated and includes additional thematic aspects.

The Street Tree Layer (STL) of the Urban Atlas (EU Copernicus project [32]) was our main source of tree cover data. Despite the name, it actually captures nearly all trees

in urban sites outside of dedicated forest areas. STL is limited to cities within so-called functional urban areas. In addition, we made use of the Urban Green Raster dataset (UGR) [33] to capture trees that (in our experience) are missed by STL. Forest areas from the LBM-DE dataset were considered in our model by inserting an additional soil cover class "forest" (see Section 2.3.3).

Finally, population data were added to the model at a reasonable resolution to be able to differentiate the number of city residents living close to UGI from those living at some distance from any green space and thus likely to suffer more from the urban heat island effect. We used population raster data (at resolution 100 m) representing population estimates in Germany from 2011.

## 2.2. Methodology

In addition to adapting the CCA model (Section 2.2.1 [22]) by introducing a forest class, population data were integrated to determine the demand for cooling capacity (Section 2.2.2), thereby creating a comprehensive ecosystem service indicator that connects supply and demand as service flow.

### 2.2.1. Assessing Cooling Capacity

The CCA approach [22] considers three main factors for the UGI cooling effect, namely: (i) shading; (ii) opportunities for evaporation; and (iii) the effective size of the green area. Shading (i) is derived from the tree canopy cover since there is a linear relationship between tree cover and the extent of shading [37]. The percentage of tree coverage is subdivided into five classes (coverage of 0–20%, >20–40%, >40–60%, >60–80%, >80–100%). For evaporation (ii), there are two essential sub-processes: plant transpiration and (pure) evaporation from water surfaces, wetlands or moisture soils. Together we call these processes evapotranspiration. The first sub-process depends on tree leaf area, for which tree cover can be taken as a proxy, as in the case of shading [10,13]. The second sub-process depends on the type of soil cover, which can be derived from land cover types. The CCA approach takes five classes of soil cover as the basis for calculating the evapotranspiration rate [22], as shown in Table 2. Finally, the effective size of green area (iii) influences the cooling effect, in particular with respect to neighbourhood effects, which show a non-linear threshold at about 2 ha [38]. Thus, in general, significantly higher values must be assumed for areas larger than 2 ha, which is considered in the fourth column of Table 2. Combining these three factors (i–iii), the basic CCA framework of [22] delivers cooling values in the range of 11–100 for each combination of tree cover and green area size while differentiating between three European climate zones. This question of climate zone can be disregarded for the proposed indicator as Germany lies in only one such zone. Therefore, we only used the values of the Atlantic climate zone for our model, which is related to Köppen's Cfb climate; cf. [13,39,40].

Because of different weather situations (wind, cloudiness, wetness, time of day, etc.), there is no exact relationship between the cooling value points (Table 2) and measured air temperature. To obtain a rough idea of the temperature changes, [22] estimates that the cooling effect for the Atlantic climate zone is less than 1 Kelvin for areas with values between 0–20, more than 2 Kelvin in areas with values above 60, and more than 3 Kelvin for areas with more than 80 value points.

**Table 2.** Values of the cooling capacity (11–100) after [22]: 230, adapted by the authors to include areas of forest (see Section 2.3.3).

| Tree Cover [1] | Soil Cover Type [2] | Cooling Value Points (Atlantic Climate) | |
|---|---|---|---|
| | | Soil Cover Type Area < 2 ha | Soil Cover Type Area ≥ 2 ha |
| ≤20% | Sealed (impervious surfaces) | 11 | 20 |
| | Bare soil | 18 | 65 |
| | Heterogeneous cover | 19 | 68 |
| | Grass (low vegetation) | 19 | 68 |
| | Water surface | 20 | 75 |
| ≤40% | Sealed (impervious surfaces) | 22 | 40 |
| | Bare soil | 27 | 74 |
| | Heterogeneous cover | 28 | 76 |
| | Grass (low vegetation) | 28 | 78 |
| | Water surface | 28 | 81 |
| ≤60% | Sealed (impervious surfaces) | 29 | 60 |
| | Bare soil | 33 | 83 |
| | Heterogeneous cover | 36 | 84 |
| | Grass (low vegetation) | 37 | 85 |
| | Water surface | 37 | 87 |
| ≤80% | Sealed (impervious surfaces) | 37 | 80 |
| | Bare soil | 44 | 91 |
| | Heterogeneous cover | 46 | 92 |
| | Grass (low vegetation) | 46 | 93 |
| | Water surface | 46 | 94 |
| ≤100% | Sealed (impervious surfaces) | 55 | 100 |
| | Bare soil | 55 | 100 |
| | Heterogeneous cover | 55 | 100 |
| | Grass (low vegetation) | 55 | 100 |
| | Water surface | 55 | 100 |
| | *Forest* | *55* | *100* |

[1] Tree cover type was determined from LBM-DE, Street Tree Layer and Urban Green Raster Germany, see Section 2.3.4 [2] Soil cover type was determined from LBM-DE, see Section 2.3.3.

2.2.2. Supply and Demand of the ES "Local Climate Regulation in Cities"

We developed an indicator to capture the effectiveness of the ecosystem service of "local climate regulation in cities" in mitigating the heat stress in city centres. Its most important parameters are:

- The urban green infrastructure (UGI), which has a high potential for reducing heat stress by its natural cooling capacity;
- The proportion of inhabitants that can benefit from UGI cooling potential near their homes, workplaces or other areas where they frequently congregate.

The latter parameter also depends on the spatial proximity of UGI with a positive climate impact (here regarded as ES supply) to residential neighbourhoods with dwellers who can benefit from cooling (shown here as ES demand) (Figure 1). Such an indicator can be used to identify those urban green spaces with a significant potential for reducing heat stress and locations where there is a higher demand for climate regulation. In this way, it is possible to assess where and through which changes more city dwellers could benefit from the cooling capacity of green infrastructure in the future.

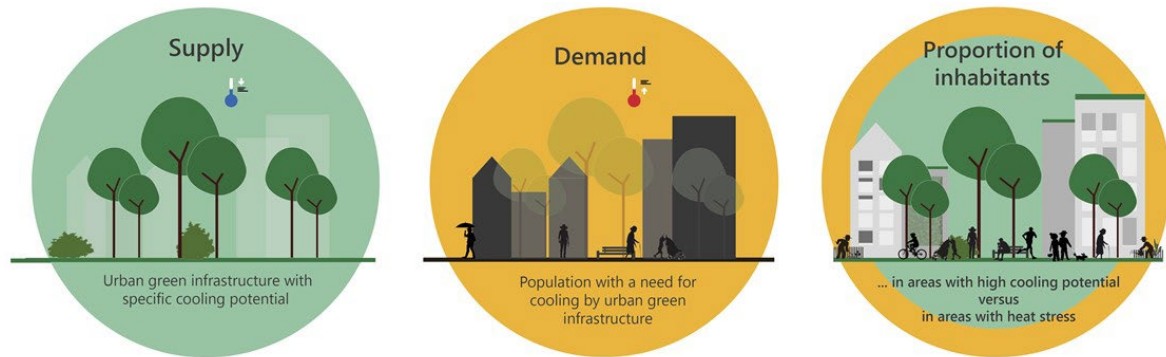

**Figure 1.** Spatial match and service flow between the supply of and demand for local climate regulation in cities.

*2.3. Application of Original CCA Methodology*

2.3.1. Basic Methodology

The main features of the original CCA methodology have already been described in Section 2.2.1. Here we explain the extensions and adaptations made for the German national indicator. For more detailed information on the original CCA methodology, see [22]. The indicator is calculated using the GIS software ArcGIS Pro 2.8.3 and the requisite Python 3.6 program script is written using the PyCharm 2019.3.1 development environment to enable a regular and comparable indicator calculation.

2.3.2. Identification of Cities ≥ 50,000 Inhabitants

The outlines of the administrative areas were used to delineate the areas of interest. Here all selected cities have at least 50,000 inhabitants and lie within an FUA [32]. Our final selection encompassed 165 cities. Some cities with more than 50,000 inhabitants were ignored because they are located outside an FUA and thus no STL data were available. These 26 cities are in are in the federal states of North Rhine-Westphalia and Baden-Württemberg, such as Gutersloh or Baden-Baden.

2.3.3. Assigning Land Cover Classes to Soil Cover Types

Following the CCA methodology [22], we classified the CLC classes into soil cover types which differ considerably in their rates of evapotranspiration: sealed, bare soil, heterogeneous cover, grass, water and forest (Table 3). Forest areas from the LBM-DE dataset were considered in our model by inserting an additional soil cover class "forest", which was not included in the original CCA model.

**Table 3.** Assignment of the LBM-DE (CLC) land cover classes to the CCA soil cover types. Adaptation by the authors in italics.

| CCA Soil Cover Types | LBM-DE (CLC) Land Cover Classes | Description |
|---|---|---|
| Sealed (impervious surfaces) | 111, 121, 122, 123, 124 | Urban settlement, industry, and transportation areas |
| Bare soil | 131, 211, 331, 332, 333, 334 | Mining, arable land, rocky areas, beaches |
| Heterogeneous cover (mixed cover of bare soil and shrubs, typical of gardens, inner courts or vacant lots) | 112, 132, 133, 142 | Non-continuous residential areas, deposits, construction, sports and leisure facilities |
| Grass (low vegetation) | 141, 221, 222, 231, 321, 322, 333, 324 | Green urban areas, vineyards, orchards, grasslands, moors and heathland, sparse vegetation, transitional woodland-shrub |
| Water | 411, 412, 421, 423, 511, 512, 521, 522, 523 | Marshes, peat bogs, intertidal flats, water courses, water bodies, lagoons, sea |
| *Forest* | *311, 312, 313* | *Broad-leaved, coniferous and mixed forests* |

### 2.3.4. Consideration of Urban Tree Cover

Tree cover was derived from two separate data sources, namely the Street Tree Layer (STL) of Urban Atlas [32] and the Urban Green Raster Germany [33]. In order to combine both of these datasets, the first was rasterised to the same resolution as the second (10 m × 10 m). For the overlay, tree information of the UGR (broad-leaf tree and needle tree) was combined with the STL (unspecified tree types). The UGR mixed type "vegetation + building area" was corrected to 100% tree cover in those areas where it intersected with the STL. The rest of the UGR mixed-type raster cells were assigned to 18% tree coverage, using calibration data of green volume for five German cities: Potsdam, Dresden, Leipzig, Hanau and Bielefeld [34,35]. The total tree cover (Figure 2, above) was added as a new attribute to each polygon in the LBM-DE dataset. Figure 2 (below) shows a satellite image so that the reader can compare the visual amount of green and the tree cover indicator calculated from the imput datasets.

### 2.3.5. Size of the Soil Cover Types

Since cooling by evapotranspiration is more effective when the soil cover type area surpasses a critical threshold of 2 ha, we assessed the size of each soil type polygon. First, neighbouring soil cover polygons of the same type (excluding "sealed") were added together (GIS function dissolve) to calculate the total area of all adjacent areas with the same soil cover type. Second, depending on these totals, the areas below 2 ha were assigned lower cooling capacity values than areas above 2 ha in size (see Table 2).

### *2.4. Model Adaptations*
### 2.4.1. Identifying the Neighbourhood around Cooling Green Areas

As urban housing is rarely constructed directly inside urban green areas, the real benefits of cooling are only enjoyed by homes located near to UGI. For our purposes, we considered that relatively large areas of UGI with high tree cover such as parks have a considerable potential to cool neighbouring residential areas [41]. In particular, we assumed that UGI with a "very high" cooling capacity above 80 points will have a cooling effect on adjacent built-up areas within a distance of 100 m (GIS function buffer) and overlaid this potential area of cooling with raster data of the urban population. Jaganmohan et al. [41] found cooling effects on neighbourhoods extending on average 110 m around parks, or 190 m around forests. The current authors decided to pick a buffer distance of 100 m, taking account of the fact that the cooling distance can be limited by barriers such as buildings. This buffer distance also compensates for geometric inaccuracies that occur when different datasets are combined. The authors of [41] identified distance cooling effects of between 0.3 and 0.7 Kelvin, corresponding to around 20 cooling capacity points (see Table 2). Thus, in our model, the cooling capacity of the areas lying in the 100 m potential cooling zone was increased by 20 cooling capacity points.

### 2.4.2. Integration of Population Data

Since ecosystem services are the contributions of nature to human well-being [42], it is important to consider not just the cooling effect of UGI but also the resulting benefit for the urban population. The final part of the indicator calculation considers the demand side of the socio-ecological system by identifying areas particularly in need of heat stress reduction, estimated by the percentage of inhabitants. The demand is considered to be the number of city inhabitants. The final ecosystem services indicator of local climate regulation in cities by green infrastructure is the percentage of these city inhabitants whose homes are in the proximity of green urban infrastructure with good cooling capacity, compared to the total city population (Figure 3). In quantitative terms, the share of persons living in areas from 61 to 100 cooling capacity value points was calculated. So, this measure as a comparison of supply (cooling capacity) and demand (city inhabitants) represents an ecosystem service flow indicator.

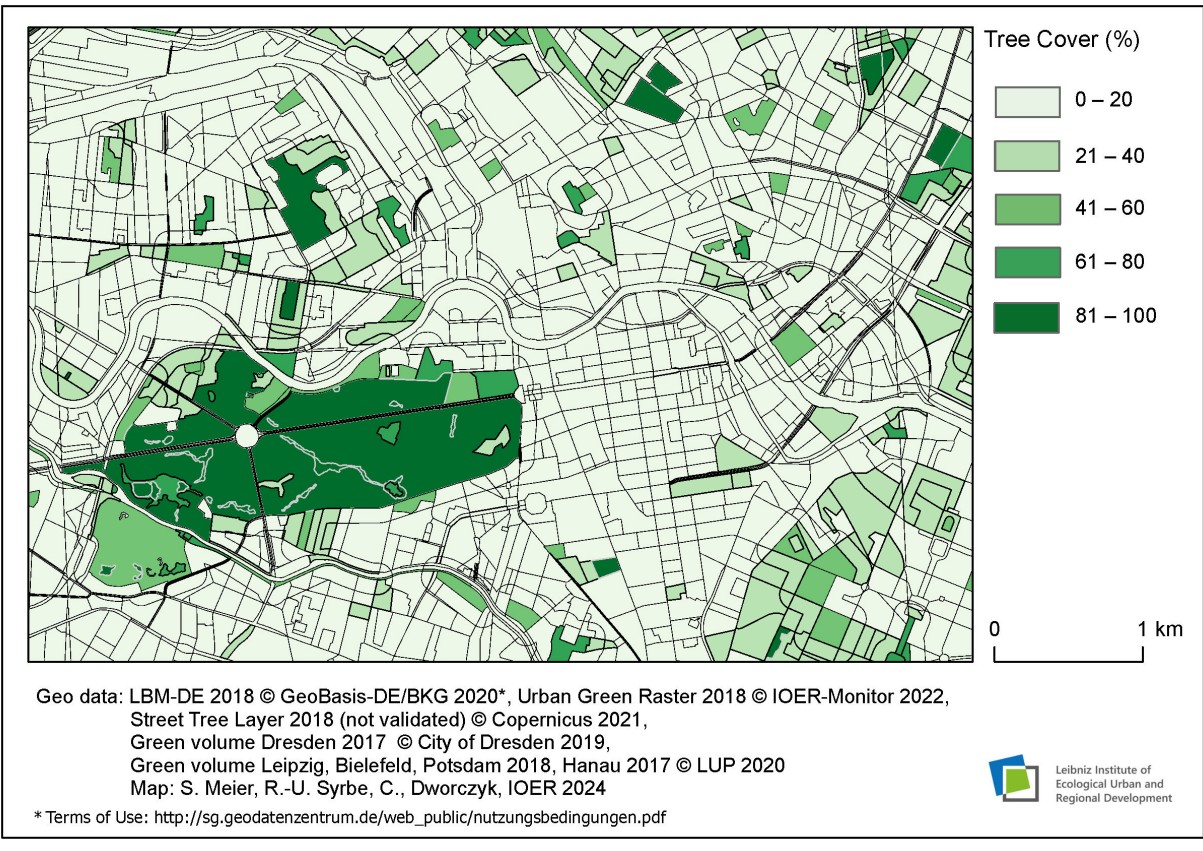

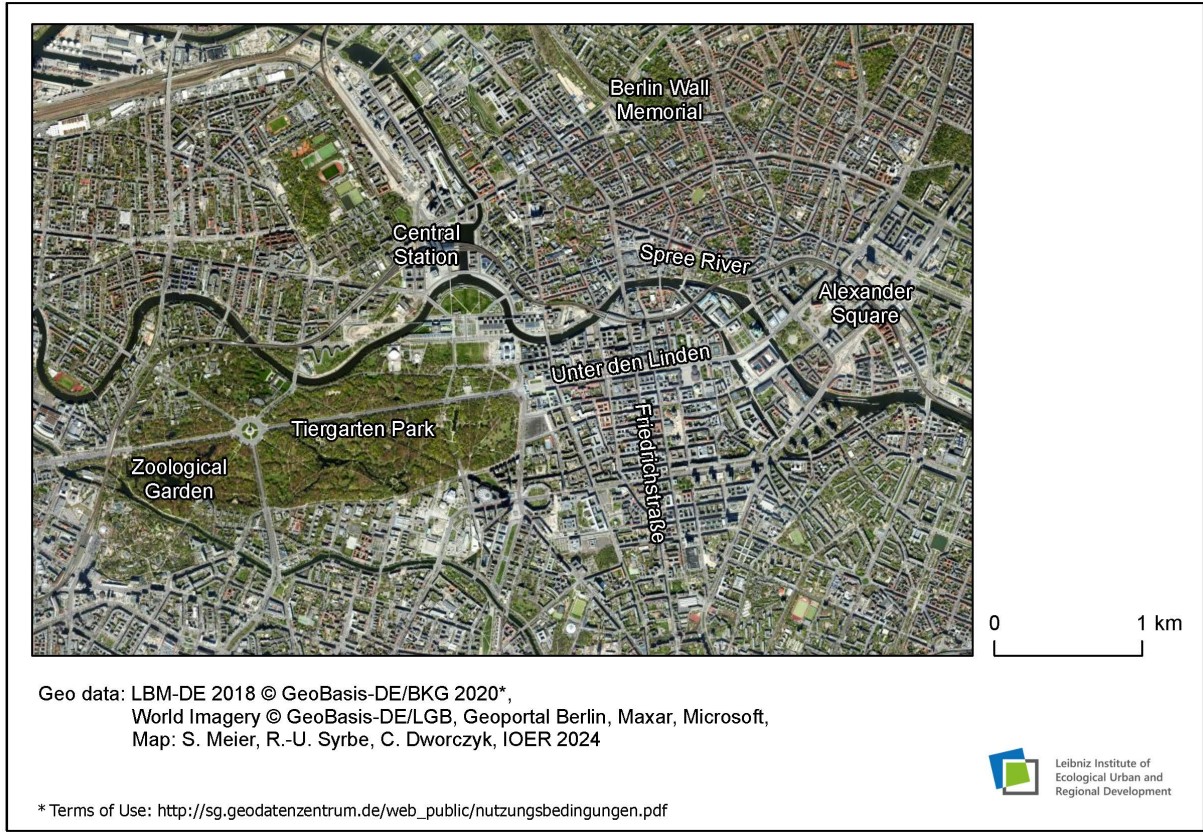

**Figure 2.** Tree cover in Berlin city centre (**above**) along with a satellite image of the same area for comparison (**below**).

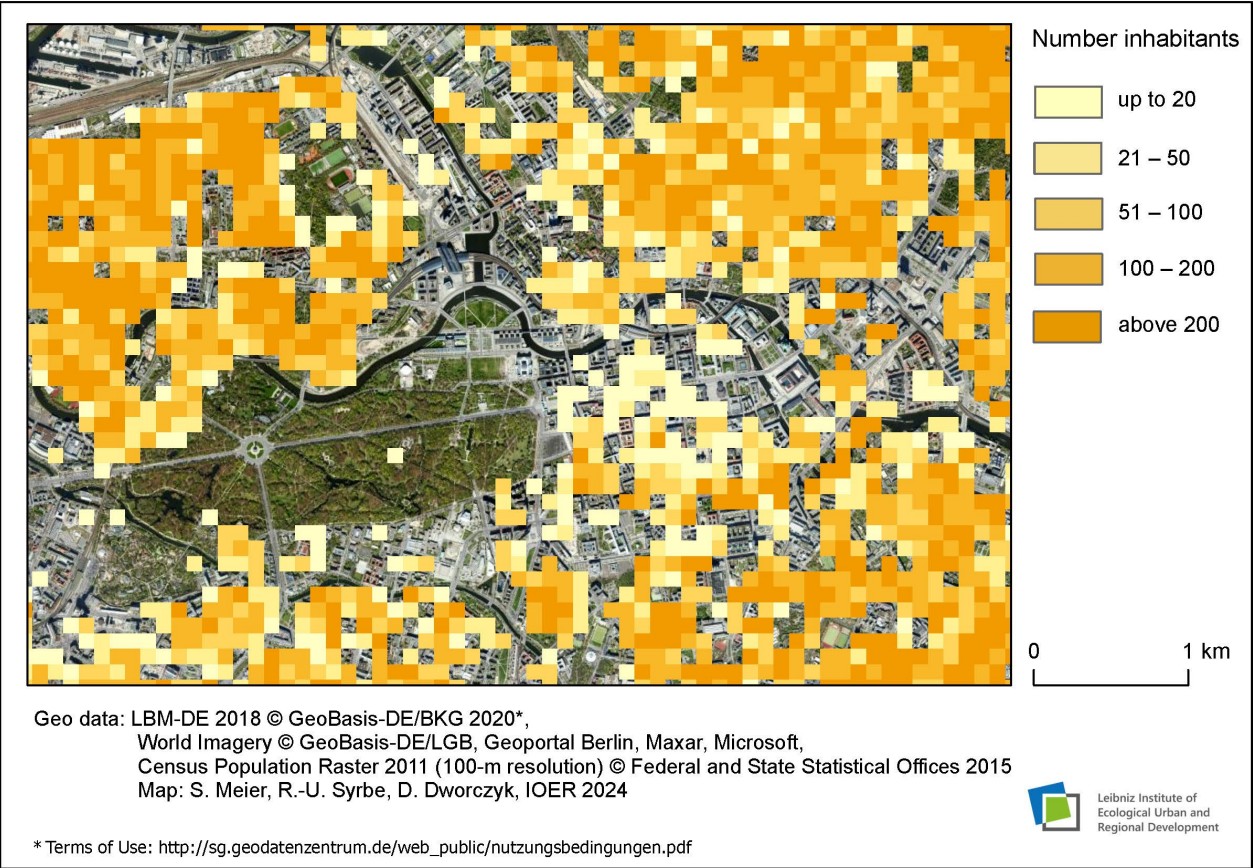

**Figure 3.** Berlin city centre showing the number of inhabitants as a proxy for the demand for cooling.

## 3. Results

### 3.1. Cooling Capacity at Local Level

The basic assessment was used to generate fine-scale maps of cooling capacity in all studied cities. One example is given in Figure 4, namely the detailed map of central Berlin, showing how the cooling effects of green infrastructure can be precisely identified at the neighbourhood level. It is easy to recognise the densely built-up centre with its low cooling capacity as well as the large green zone to the west of the centre, which is the famous "Tiergarten" park. The buffer areas around the park and other urban green spaces indicate areas where local residents are likely to benefit from the proximity cooling effect of nearby UGI.

The German capital benefits from the Spree river, which provides cooling as it meanders through the centre, even though trees are rather sparse along its banks. Overall, Berlin is a relatively green city, with areas of good cooling potential between various densely built-up, compact centres. Since the main aim here is to develop a nationally comparable indicator set, the average cooling capacity as physical supply is calculated for each city of interest.

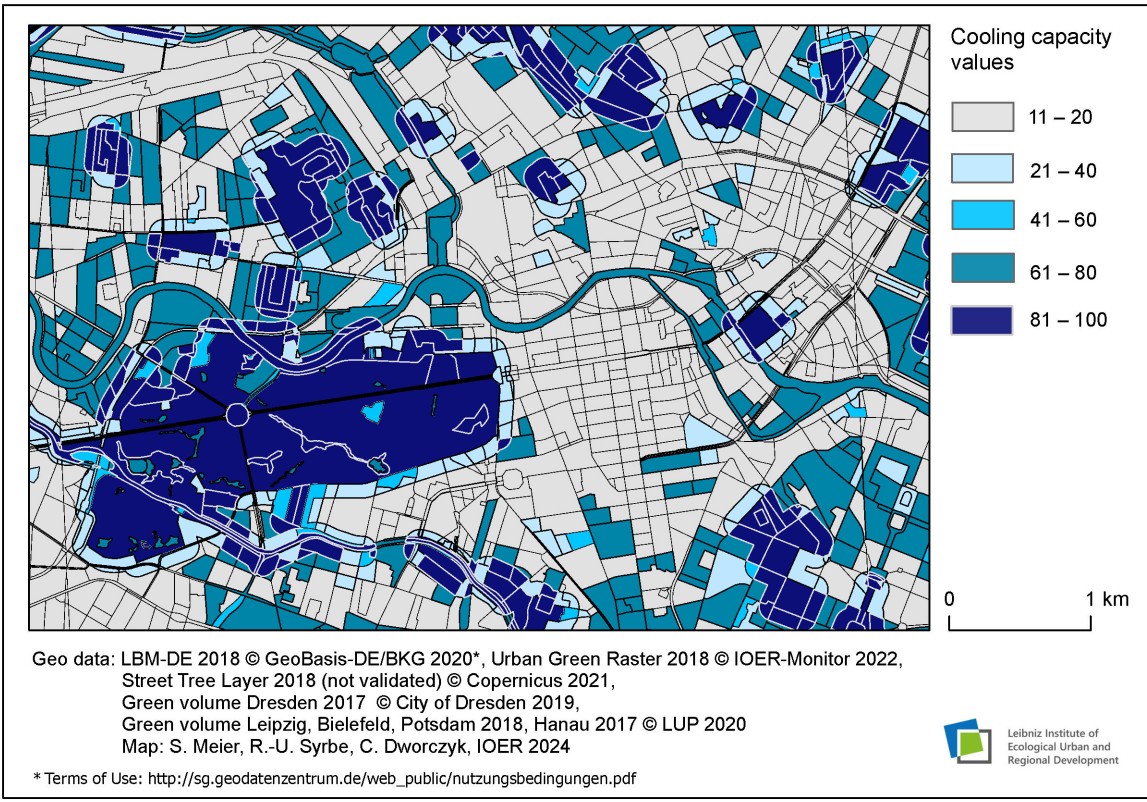

Geo data: LBM-DE 2018 © GeoBasis-DE/BKG 2020*, Urban Green Raster 2018 © IOER-Monitor 2022,
Street Tree Layer 2018 (not validated) © Copernicus 2021,
Green volume Dresden 2017 © City of Dresden 2019,
Green volume Leipzig, Bielefeld, Potsdam 2018, Hanau 2017 © LUP 2020
Map: S. Meier, R.-U. Syrbe, C. Dworczyk, IOER 2024

* Terms of Use: http://sg.geodatenzentrum.de/web_public/nutzungsbedingungen.pdf

**Figure 4.** Detailed map of Berlin city centre showing the cooling capacity of UGI as value points ranging from grey (no cooling effect, risk of urban heat islands forming) to varying shades of blue up to dark blue (highest cooling capacity) (note that the number of local inhabitants is neglected here).

## 3.2. Cooling Capacity at National Level

As our aim was to conduct an assessment at national scale, we calculated the mean cooling capacity values for all selected cities (Figure 5). The resulting dataset in its physical sense can be seen as an indicator for the supply of an ecosystem service. The map considers the land use and tree cover of green infrastructure, the individual sizes of green spaces and the buffer zone (100 m) around large green areas with high cooling capacities [43]. Cities with large areas of green infrastructure (especially forests and water bodies) within their administrative boundaries, such as Potsdam, Jena, Saarbrucken and Trier, as well as several cities in the Ruhr area and the south-west of Germany, have high cooling capacity; this contrasts with more compact cities as well as those where green space is largely at the city periphery, such as Magdeburg, Halle and Nuremberg, which all have a lower cooling capacity. The average values range from 57 in Ludwigshafen/Rhine to a maximum of 88 points in Siegen. The map of all German indicator values is shown in Figure 5.

## 3.3. Ecosystem Service Indicator "Local Climate Regulation in Cities"

The map in Figure 6 shows the final values of the indicator "Local climate regulation in cities" at the national level for the cities of interest. It shows the proportion of the population provided with (at least) a good cooling effect (more than 61 value points) by overlaying assessed green spaces, their 100 m buffer areas and the number of inhabitants. The overall results range from 47% (Nuremberg) to 93% (Meerbusch). In addition to the pure cooling effect of Figure 5, the spatial distribution of high-quality and sufficiently large green spaces is crucial for meeting the demand of urban residents. Clearly, it is essential to the delivery of this ES that the most densely populated areas are located in the vicinity (max. 100 m distance) of highly climate-effective green spaces.

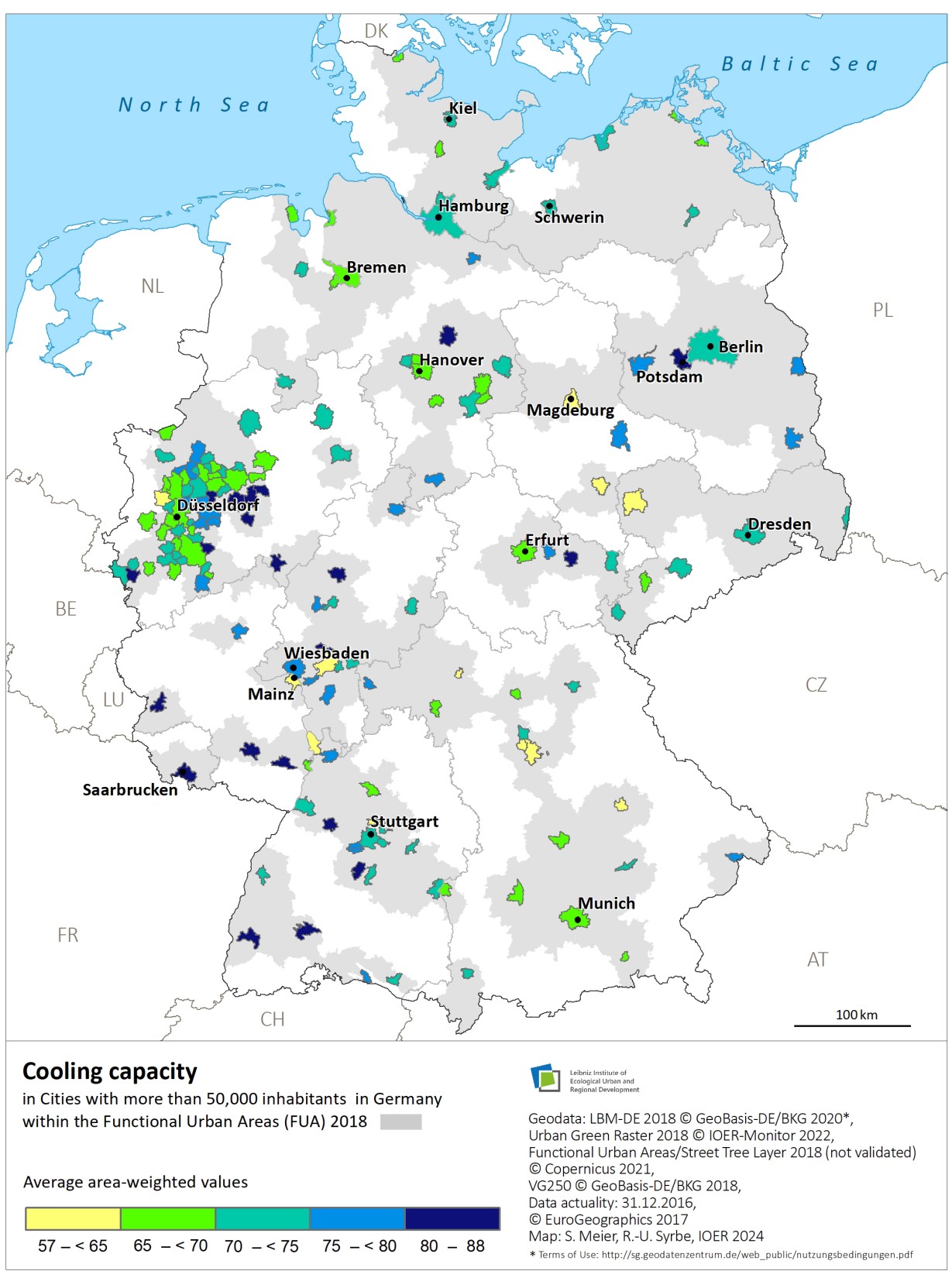

**Figure 5.** Average area-weighted cooling capacity values for 165 German cities with populations over 50,000 within the functional urban areas (grey areas) of the Urban Atlas.

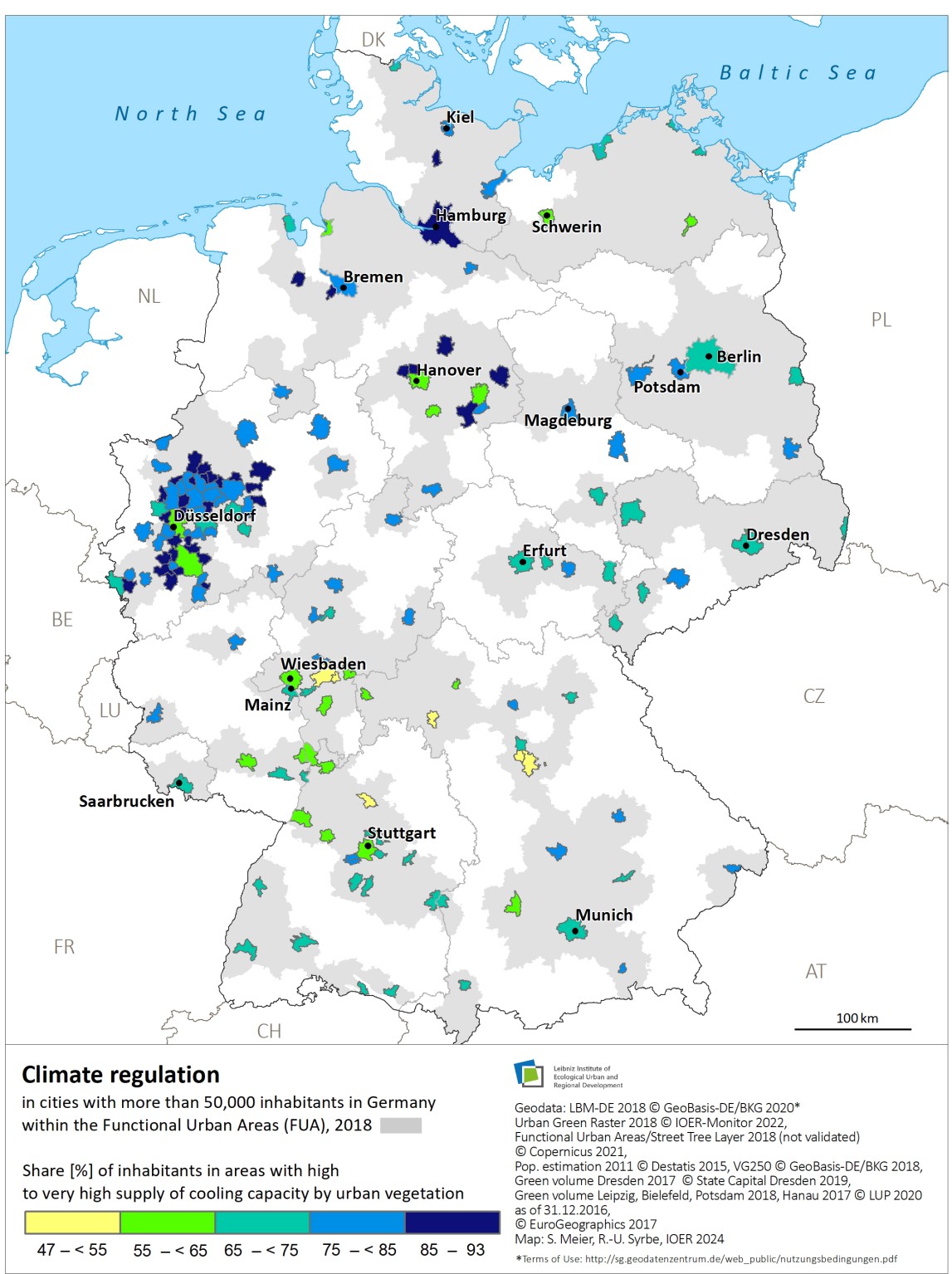

**Figure 6.** Proportion of the cities' population provided with a cooling capacity from 61 to 100 value points in 165 German cities with over 50,000 inhabitants within the functional urban areas (grey).

The map in Figure 6 shows that more than 85% of the population in 37 of the 165 analysed cities are provided with good to very good cooling capacity by green infrastructure, such as Hamburg, Wolfsburg, Celle and Recklinghausen. The next best class of climate regulation provision, i.e., with more than 75% of inhabitants enjoying good or very good cooling capacity, is found in 56 cities, such as Bonn, Kiel and Potsdam. In contrast, we

found only six cities in which less than 55% of the population are likely to benefit from the cooling effects of urban greenery.

## 4. Discussion

The aim of this paper was to provide a methodology to assess the cooling benefits of urban green infrastructure across different urban settings. This includes an assessment of the physical cooling capacity of UGI and the demand for local climate regulation by the urban population.

The results at the national level enable comparisons between cities. At the local level, the small-scale maps allow the analysis of individual urban green infrastructure elements and their impact on climate regulation, which can reveal and identify deficiencies within the analysed cities [44]. Under the developed method, it is possible to regularly monitor land use changes and observe if urban development is improving or restricting the supply of the ecosystem service. The methodology can support measures to upgrade urban green infrastructure even when space is limited, for example by unsealing soil, planting new trees and enlarging the extent of vegetative cover (to increase the number of green spaces larger than two hectares). The detailed map shows the extent of possible improvements with the option to zoom into the map to examine individual urban structure units (Figures 2–4).

Even though our focus here was solely on daytime cooling in large cities, we were forced to neglect some relevant factors. In particular, it can be assumed that over the course of the day, some urban residents will change their location, whether to go to school or work or to go shopping. Such population movements will also impact the demand for heat stress reduction. Yet the model presented here does not capture the specific distribution of the population during the day, i.e., temporary increases in population density in certain areas of a city (due, for instance, to tourist groups) are ignored. On the other hand, there is some potential for a finer-scaled population analysis: currently, population figures are calculated on the basis of the national census for grid areas of 100m × 100 m, which is a relatively coarse spatial resolution compared to the other data used.

In general, it is a disadvantage to rely on data with different timeliness, spatial resolutions and update rates. The presented indicator also has several additional limitations that should be addressed for future developments. First, the assessment of cooling capacity in our approach is designed for several European climate zones. While this could be refined to reflect more specific local climates within Germany, such climate zones are difficult to delineate. Second, the assessed impact of evapotranspiration assumed an unlimited supply of water. In the event of droughts, real evaporation would be much lower than suggested by the modelled cooling capacity. Third, the effects of roof and wall greening could be incorporated into future modelling even though these largely have an impact on individual houses; the so-called heat resilience city tool [19] can do this at the local level. Fourth, the issue of distances is still critical. Even if the indicator refers spatially to the residential population, it only represents the outdoor climate situation. The cooling effects within the buildings cannot be addressed since totally different physical processes are valid. Clearly, the building construction, insulation and windows will have a significant influence on indoor temperatures. However, the temperatures within the urban fabric outside the buildings are difficult to model within such a national framework; in particular, the warming effects of sealed surfaces mentioned in the introduction by explaining the urban heat island effect could not implemented completely. Therefore, the IOER developed the so-called heat resilient city (HRC) tool [19], which explicitly addresses the effects both outside and inside the buildings.

Any ecosystem service indicator must do more than just measure or model a service. Above all, it should reduce the information content of the complex socio-ecological system between humans and their (living) environment, presenting it in an understandable way, as well as being "quantifiable, sensitive to changes in land use, temporally and spatially explicit and scalable" [23] p. 486. Furthermore, if an indicator is to work at the national



level, suitable input data must be available for this purpose. The "local climate regulation in cities" indicator presented here fulfils these requirements [24].

Generally, it is important to map supply and demand separately (cf. [45]), as the demand for a reduction in heat stress can change over time and be redistributed spatially. After their separate measurement and evaluation, the supply and demand sides can be compared. These two factors can then be merged to develop urban planning measures [46] that provide the population with high-potential green infrastructure and reduce heat stress. This requires the provision of information to assess which features are essential for good to very good cooling capacity and to identify the urban areas with particularly high population. In this way, priority neighbourhoods can be identified at the urban level which require the provision of high cooling capacity through urban greenery. In our study, the population raster data used to determine the number of inhabitants and their local densities were drawn from the last census of 2011 at a resolution of 100 by 100 m, which is the most recent available data source at this scale. However, in 2021 a new population estimate was carried out, but the relevant data have not yet been released. This upcoming population data will enable us to update not only the supply side of assessment but also the population-dependent demand side.

Particular attention should also be paid to urban areas that contain a high number of socially disadvantaged residents [47]. These areas tend to have fewer green spaces, bringing a higher risk of heat stress for local inhabitants than for those living in wealthier neighbourhoods, as has already been proven for some major German cities such as Dortmund (see [47]). Another issue regarding heat stress is the particular vulnerability of the elderly and chronically ill; the ratio of such people in an urban site could be derived in future studies by investigating the number of care/retirement homes or hospitals. Such information is still lacking in the database used here but could be implemented as soon as relevant data are made available.

For a complete assessment of the ecosystem service of "climate regulation in cities", additional aspects must be considered, such as night-time cooling, the supply of fresh air (ventilation) and all other climate regulation potentials such as global climate protection, attenuation of dry or wet periods, and protection against storms and other severe weather conditions. Additional indicators such as the land use and land use change indicators of national climate reporting are able to reflect these factors, e.g., [48].

Ecosystem services are foreseen to be integrated into environmental economic accounting. Eurostat has suggested indicators for assessing local climate regulation in urban areas to facilitate this integration [49]. The proposed indicators, adapted from Marando et al. [50], also assess local climate regulation in urban areas across FUAs, utilising land use and land cover data alongside tree cover density. However, these models require climate-related data, including satellite data on land surface temperatures and information on evapotranspiration rates, to accurately quantify cooling effects. In comparison, the CCA methodology applied here enables a faster assessment at the national level.

According to Eurostat, it is mandatory to report the reduction in heat exposure, expressed as the average temperature reduction on days when maximum temperatures exceed 25 °C, specifically for the ecosystem type "settlement and other artificial areas" within the functional urban area (FUA). The demand for heat-reducing ecosystem services is determined by the number of days surpassing 25 °C. Consideration of other temperature maxima and ecosystem types is voluntary [49]. The presented framework additionally considers population data, which could be enhanced by other demographic information, enabling deeper analyses of vulnerable groups such as children and the elderly [51].

Decision-makers should set national goals to avoid a deterioration in natural climate regulation despite the unavoidable densification of cities [25]. This is particularly relevant in cities where the majority of the population is already supplied with high cooling capacity values (61 to 100). On the other hand, there is a justified need for action to improve climate regulation, particularly in urban neighbourhoods with high population densities and in cities where a large share of the population has little potential to reduce heat stress [52]. Here

strategies of dual internal development can be useful, in which space-saving construction is harmonised with the promotion and upgrading of urban green spaces (see [53–55]).

## 5. Conclusions

The paper presents a new indicator in the field of ecosystem conditions and services that captures local climate regulation in cities for the whole of Germany. It enables a nation-wide comparable and regularly updated assessment of the climate regulation performance of urban green infrastructure. Thanks to the relatively small-scale assessment at the level of individual urban neighbourhoods, it is possible to quickly identify the need for action, and later to monitor the improvement or deterioration in people's well-being. Furthermore, individual cities can be compared, thereby motivating municipal authorities to speed up their efforts to green residential neighbourhoods and to reduce the incidence and severity of urban heat islands. By comparing supply (of cooling) with demand (population density), it becomes easier to identify important ecosystems based on their proximity to housing, so that planning (and conserving) measures for UGI can also be intelligently selected. The authors intend for the evaluation results to be made regularly available to all potential users, e.g., in planning and urban development.

Based on the guiding principle of "dual internal development" [54], the presented national ecosystem service indicator can pinpoint potential conflicts in land use as well as opportunities for win–win situations. For example, measures such as connecting individual elements of green infrastructure or ensuring the proximity of green spaces to residential buildings can help reduce heat stress in densely built-up urban areas. Our indicator can make a useful contribution to the EU Biodiversity Strategy for 2030, which aims for the "promotion of healthy ecosystems, green infrastructure and nature-based solutions" to be "systematically integrated into urban planning, including in the planning of public spaces and infrastructure and in the design of buildings and their surroundings" ([42] p. 15).

**Author Contributions:** Conceptualisation—K.G., M.M. and R.-U.S.; methodology—M.M., R.-U.S. and S.M.; software—S.M. and R.-U.S.; validation—R.-U.S. and S.M.; formal analysis—K.G.; investigation—M.M., R.-U.S. and C.D.; resources—S.M.; data curation—M.M. and S.M.; writing: original draft preparation—R.-U.S. and M.M.; writing: review and editing—S.M., K.G. and C.D.; visualisation—S.M. and R.-U.S.; supervision—K.G.; project administration—K.G.; funding acquisition—K.G. All authors have read and agreed to the published version of the manuscript.

**Funding:** This research was funded by the Federal Agency for Nature Conservation as part of the project "Further development of the nationwide indicator set for ecosystem services" (funding code ID3518810400).

**Data Availability Statement:** The data developed in this research are provided under the international F.A.I.R. conditions and ODC-By licence in the IOER research data centre: https://ioer-fdz.de/en/germanys-ecosystems, accessed on 6 May 2024, both in the section Ecosystem Conditions: Urban ecosystems/Cooling effect of green infrastructure and in the section Regulating Services/Climate regulation in cities.

**Acknowledgments:** The team would particularly like to thank Burkhard Schweppe-Kraft and Beyhan Ekinci for initiating and supervising the project. The origin of the project was Michelle Moyzes' Master's thesis "Development of an indicator to assess the ecosystem service 'climate regulation in cities'". Our special thanks go to Astrid Ziemann and Uta Moderow from the Chair of Meteorology at TU Dresden for their expert supervision of the Master's thesis, to Kerstin Ludewig for designing Figure 1 and to Derek Henderson for language polishing.

**Conflicts of Interest:** Michelle Moyzes is employed by Energiedienst Holding AG. The authors declare no conflicts of interest.

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
