# Peer review of "Assessment and Monitoring of Local Climate Regulation in Cities by Green Infrastructure—A National Ecosystem Service Indicator for Germany"

_land, doi:10.3390/land13050689_

Round 1

Reviewer 1 Report

Comments and Suggestions for Authors

The cooling effect within the building development is certainly ideally overrated, as the influence of the development structure is a very decisive factor. This is certainly a point that should be included in the considerations of this work for the future.

The description of the method and its derivation could be explained in more detail in the text. 

Author Response

Reviewer 1

Comments and Suggestions for Authors

The cooling effect within the building development is certainly ideally overrated, as the influence of the development structure is a very decisive factor. This is certainly a point that should be included in the considerations of this work for the future.

Answer: This issue is explained in more detail in discussion now, see new lines 386-394.

The description of the method and its derivation could be explained in more detail in the text. 

Answer: The description of the method and especially of its derivation has been explained in more detail following the reviewer 2 suggestions, see new lines 173-175, 226, 260-261, 277-284.

Reviewer 2 Report

Comments and Suggestions for Authors

This paper offers an approach to quantify urban climate regulation ecosystem services provided by green spaces. Although highly simplified, the approach is novel and useful in the indicators use of national-scale data sets and outcomes. In doing so, the research illustrates the widespread benefits provided by UGI.

The paper is well written, with useful illustrations demonstrating the analysis for Berlin and at the national level. Some specific recommendations follow:

Line 114: insert “Climate” before Cooling Assessment.

Table 1: not sure why city names is listed as a data source. Is spatial scale needed for FUA data?(understand that these data is used differently in the analysis, but the data certainly has a mapping scale.)

Table 2: I wished for an easier link between tables 1 and 2: e.g., STL/UGR was used for the tree cover data,  LBM – DE was used for the soil type…

Line 223 – my impression is that this sentence is missing a “not” before “included in the original CCA model.”

Line 242 - Recommend change “green area” header in table 2 to “soil cover type area” to match this paragraph, or (preferably) change soil cover area in this paragraph to green area. In any case, the terminology should be consistent.

Line 256 – delete “and an area of 2 ha size.” Clause not necessary. Only areas over 2 ha have cooling capacities above 80 points according to table 2.

Line 274 - “(compared to the total population).” total population of what? the city? Is this meant to be the description for the calculation of the average cooling capacity of each case study city? Confusing as written. More explanation is needed about what is meant by “the value.”

Figure 6: “good to very good cooling capacity” means cooling capacity value greater than 80? This scale could be more clearly defined in the figure caption and in the methods.

Similarly, on line 433, the Discussion mentions "high cooling capacity values of 61 to 100." The scale should be more prominently provided earlier and throughout the paper.

Discussion: while appreciate the minimal inputs and ability to conduct the analysis at scale, it would be useful to expand further upon the simplifying limitations and their implications - as an example, nighttime warming is briefly mentioned, but the neighborhood warming effects of areas with “sealed” cover types could be explicitly addressed.

Line 397: yes, this struck me immediately in reading the methods that the 2011 census data set is out of sync with the rest of the data. If choosing not to use this better data source, it would be worth considering the implications in more detail. Were there some cities that changed a lot between those two data periods? What was the overall change? What kind of change?

Author Response

Reviewer 2

Comments and Suggestions for Authors

This paper offers an approach to quantify urban climate regulation ecosystem services provided by green spaces. Although highly simplified, the approach is novel and useful in the indicators use of national-scale data sets and outcomes. In doing so, the research illustrates the widespread benefits provided by UGI.

Answer: Thank you very much.

The paper is well written, with useful illustrations demonstrating the analysis for Berlin and at the national level. Some specific recommendations follow:

Line 114: insert “Climate” before Cooling Assessment.

Answer: This has been done.

Table 1: not sure why city names is listed as a data source. Is spatial scale needed for FUA data?(understand that these data is used differently in the analysis, but the data certainly has a mapping scale.)

Answer: The name is deleted.

Table 2: I wished for an easier link between tables 1 and 2: e.g., STL/UGR was used for the tree cover data,  LBM – DE was used for the soil type…

Answer: I added 2 footnotes to the table to clarify these links, see new lines 173-175.

Line 223 – my impression is that this sentence is missing a “not” before “included in the original CCA model.”

Answer: I added the indeed missing “not”.

Line 242 - Recommend change “green area” header in table 2 to “soil cover type area” to match this paragraph, or (preferably) change soil cover area in this paragraph to green area. In any case, the terminology should be consistent.

Answer: This has been done.

Line 256 – delete “and an area of 2 ha size.” Clause not necessary. Only areas over 2 ha have cooling capacities above 80 points according to table 2.

Answer: This is deleted.

Line 274 - “(compared to the total population).” total population of what? the city? Is this meant to be the description for the calculation of the average cooling capacity of each case study city? Confusing as written. More explanation is needed about what is meant by “the value.”

Answer: Some sentences are added to explain this in more detail and to clarify the information, see the new lines 277-284.

Figure 6: “good to very good cooling capacity” means cooling capacity value greater than 80? This scale could be more clearly defined in the figure caption and in the methods.

Answer: No, it is meant over 61 value points, I added this in the capture. See the new lines 344-346.

Similarly, on line 433, the Discussion mentions "high cooling capacity values of 61 to 100." The scale should be more prominently provided earlier and throughout the paper.

Answer: Yes, this mentioned similarly throughout the paper now, see above.

Discussion: while appreciate the minimal inputs and ability to conduct the analysis at scale, it would be useful to expand further upon the simplifying limitations and their implications - as an example, nighttime warming is briefly mentioned, but the neighborhood warming effects of areas with “sealed” cover types could be explicitly addressed.

Answer: This issue is explained in more detail in discussion now, see new lines 386-394.

Line 397: yes, this struck me immediately in reading the methods that the 2011 census data set is out of sync with the rest of the data. If choosing not to use this better data source, it would be worth considering the implications in more detail. Were there some cities that changed a lot between those two data periods? What was the overall change? What kind of change?

Answer: The new census data is not yet available, a clarification is added; thus, comparisons can not be made now, see new lines 412-415.
